# Immunogenic and Protective Properties of Recombinant Hemagglutinin of Influenza A (H5N8) Virus

**DOI:** 10.3390/vaccines12020143

**Published:** 2024-01-29

**Authors:** Nadezhda B. Rudometova, Anastasia A. Fando, Lyubov A. Kisakova, Denis N. Kisakov, Mariya B. Borgoyakova, Victoria R. Litvinova, Vladimir A. Yakovlev, Elena V. Tigeeva, Danil I. Vahitov, Sergey V. Sharabrin, Dmitriy N. Shcherbakov, Veronika I. Evseenko, Ksenia I. Ivanova, Andrei S. Gudymo, Tatiana N. Ilyicheva, Vasiliy Yu. Marchenko, Alexander A. Ilyichev, Andrey P. Rudometov, Larisa I. Karpenko

**Affiliations:** 1Federal Budgetary Research Institution State Research Center of Virology and Biotechnology «Vector», Rospotrebnadzor, Koltsovo 630559, Novosibirsk Region, Russiaorlova_la@vector.nsc.ru (L.A.K.); def_2003@mail.ru (D.N.K.); borgoyakova_mb@vector.nsc.ru (M.B.B.); litvinova_vr@vector.nsc.ru (V.R.L.); tigeeva_ev@vector.nsc.ru (E.V.T.); danilvahitov2001@mail.ru (D.I.V.); sharabrin_sv@vector.nsc.ru (S.V.S.); dnshcherbakov@gmail.com (D.N.S.); ivanova_ki@vector.nsc.ru (K.I.I.); gudymo_as@vector.nsc.ru (A.S.G.); ilicheva_tn@vector.nsc.ru (T.N.I.); marchenko_vyu@vector.nsc.ru (V.Y.M.); ilyichev@vector.nsc.ru (A.A.I.); rudometov_ap@vector.nsc.ru (A.P.R.); lkarpenko1@ya.ru (L.I.K.); 2Institute of Solid State Chemistry and Mechanochemistry, Siberian Branch of Russian Academy of Sciences, Novosibirsk 630090, Novosibirsk Region, Russia; evseenkov@inbox.ru

**Keywords:** influenza A (H5N8) virus, recombinant hemagglutinin, CHO-K1, immunogenicity, protection

## Abstract

In this study, we characterized recombinant hemagglutinin (HA) of influenza A (H5N8) virus produced in Chinese hamster ovary cells (CHO-K1s). Immunochemical analysis showed that the recombinant hemagglutinin was recognized by the serum of ferrets infected with influenza A (H5N8) virus, indicating that its antigenic properties were retained. Two groups of Balb/c mice were immunized with intramuscular injection of recombinant hemagglutinin or propiolactone inactivated A/Astrakhan/3212/2020 (H5N8) influenza virus. The results demonstrated that both immunogens induced a specific antibody response as determined by ELISA. Virus neutralization assay revealed that sera of immunized animals were able to neutralize A/turkey/Stavropol/320-01/2020 (H5N8) influenza virus—the average neutralizing titer was 2560. Immunization with both recombinant HA/H5 hemagglutinin and inactivated virus gave 100% protection against lethal H5N8 virus challenge. This study shows that recombinant HA (H5N8) protein may be a useful antigen candidate for developing subunit vaccines against influenza A (H5N8) virus with suitable immunogenicity and protective efficacy.

## 1. Introduction

In the 2016–2017 winter season, a highly pathogenic A (H5Nx) avian influenza virus of clade 2.3.4.4 caused one of the largest epizootics in Europe (2781 outbreaks) [1]. The years 2020–2021 brought even more outbreaks in poultry (3777) and mass deaths of wild birds [2]. During that wave, multiple reassortants involving highly pathogenic A (H5Nx) viruses, appeared and diversified rapidly, resulting in the co-circulation of 19 various genotypes that belong to five various subtypes (H5N1, H5N3, H5N4, H5N5, and H5N8) [3]. The global spread of these highly pathogenic viruses and their recombination with various local low-pathogenic viruses occurred much faster and at higher rates than has been previously observed for other strains of H5, H7, or H9 viruses, ultimately increasing the multiplicity of clade 2.3.4.4 viruses [4].

A vivid example is the outbreak of avian influenza A (H5N8), which caused massive deaths of wild and domestic birds and resulted in serious economic losses to agricultural producers. Avian influenza H5N8 clade 2.3.4.4b viruses have killed more than 33 million birds since January 2020 [5,6]. In addition, there is an increase in the number of cases of avian influenza A (H5N8) virus detection in mammals, which are biologically closer to humans than birds [7]. Hence there are concerns that the virus may adapt to infect humans more easily [8]. In 2020, the first cases of human infection with genetic reassortant HPAI A (H5N1)—H5N8 were registered in Russia; these reassortants of influenza virus require careful monitoring and comprehensive studies [7,9,10].

Due to the widespread distribution of avian influenza A (H5N8) virus in bird populations and the associated potential risks to human health, it is necessary to assure readiness to pandemic influenza at all levels, including vaccine prophylaxis [4,10,11].

The production of inactivated or split influenza vaccines is labor intensive process. It takes place in biosafety level 2+ or 3 facilities for egg-based or cell culture-based virus propagation, inactivation, and purification processes [12]. Recombinant subunit vaccines have the advantage of quick and efficient production with low biosafety requirements.

One of the main components of influenza vaccines is hemagglutinin (HA), as it induces neutralizing antibodies that stimulate protective immunity to the virus [12,13]. HA sequences from avian and human influenza viruses belong to the same subclades 2.3.4.4b (H5N1, H5N6, H5N8) and have high homology, and some differences. Most of these differences are located in the RBD region of HA1 domain (E130D, A144T, V152L, R173Q, T192I, and V214A). Mutations in this region can affect the transmissibility, pathogenicity, and antigenicity of the virus [14]. Therefore, when developing vaccines against H5N8, it seems better to use the hemagglutinin sequence of the H5N8 virus. In this work, we used the hemagglutinin sequence of the influenza virus A/turkey/Stavropol/320-01/2020 gene, which was isolated from poultry and is antigenically related to the influenza virus strain A/Astrakhan/3212/2020 (H5N8), isolated from humans and selected by WHO as a candidate vaccine strain in the event of a human outbreak [15]. Recombinant hemagglutinins of H1N1, H7N9, H5N1, and H3N2 produced in different expression systems have shown their safety and ability to induce neutralizing antibodies that protect against infection with homologous influenza viruses [12,13,16,17,18,19,20]. Since the antigenic structures of influenza HA proteins are complex and require glycosylation to maintain immunogenicity, eukaryotic cells are more suitable as producers.

In this work, we present data on obtaining a stable CHO-K1 cell lines, expressing hemagglutinin from the influenza A virus (H5N8) virus (A/turkey/Stavropol/320-01/2020), and the results of studying immunogenic and protective properties of the recombinant hemagglutinin.

## 2. Materials and Methods

### 2.1. Virus Strains, Bacteria, Cell Cultures

*E. coli* Stbl3 strain (Invitrogen, Waltham, MA, USA) was used for plasmid DNA production. Bacteria were cultivated in LB nutrient medium.

CHO-K1 cell culture was used to obtain a stable cell clones for HA production (Cell Culture Collection of FBRI SRC VB «Vector», Rospotrebnadzor, Koltsovo, Russia).

MDCK cell culture was used for influenza virus production and neutralization assay (Cell Culture Collection of FBRI SRC VB «Vector», Rospotrebnadzor). For virus production, MDCK cells were cultured in 50–250 mL flasks for 1–2 days. When a subconfluent monolayer formed, the growth medium was removed and the viral suspension added. The bottle was placed in a CO_2_ incubator at 37 °C for 30–40 min, after which the virus propagation medium was added to 1/5 of the volume of the bottle. The composition of the medium for virus production was as follows: DMEM nutrient medium with 4.5 g/L glucose, with GlutaMAX™ (Gibco, Grand Island, NY, USA); trypsin from bovine pancreas (Sigma Aldrich, Saint Louis, MO, USA) and antibiotic–antimycotic (Gibco, Grand Island, NY, USA).

A/turkey/Stavropol/320-01/2020 (H5N8) influenza virus (EPI1114749) was used in the neutralization assay and A/Astrakhan/3212/2020 (H5N8) influenza virus (EPI1846961) (FBRI SRC VB «Vector», Rospotrebnadzor) was used for immunization and in challenge experiments. These viruses were isolated in the course of monitoring the influenza virus in the Russian Federation.

A/dalmatian pelican/Astrakhan/213-2V/2022 (H5N1) influenza virus and A/chicken/Khabarovsk/24-1V/2022 (H5N1) influenza virus [21], A/gyrfalcon/Washington/41088-6/2014 (H5N8) influenza virus, and A/chicken/Vietnam/ NCVD-15A59/2015 (H5N6) influenza virus [22] were used for the hemagglutination inhibition assay.

### 2.2. Cloning and Expression of Recombinant Protein

The design of the nucleotide sequence encoding recombinant hemagglutinin of influenza A (H5N8) virus was based on the native hemagglutinin A/turkey/Stavropol/320-01/2020 gene (EPI1114749). After design, codon optimization of the gene for expression in mammalian cells was performed using the Codon Adaptation Tool (https://www.jcat.de/, accessed on 1 July 2022). Gene synthesis was performed by DNA-Synthesis (Moscow, Russia). The obtained gene was inserted into the integration plasmid vector pVL3.

A stable recombinant hemagglutinin producer was obtained from the CHO-K1 cell line by transfection of cells with the plasmids pVL3-HA/H5 and pCMV(CAT)T7-SB100 (Addgene, Watertown, MA, USA) at a 10:1 ratio, respectively, using Lipofectamin 3000 (Invitrogen). After three days, the culture medium was replaced with a fresh medium supplemented with the selective antibiotic puromycin at the concentration of 10 μg/mL (InvivoGen, San Diego, CA, USA). The pVL3-HA/H5 integration cassette included the puromycin resistance gene for, so selection of resistant cells was performed using puromycin. Highly productive clones were isolated by dilution cloning, and one of the productive clones was cultured in roller vials at 37 °C in DMEM/F-12 medium (Servicebio, Chicago, IL, USA) supplemented with 5% FBS (Himedia, Kennett Square, PA, USA).

### 2.3. Model Building

Modeling of influenza A (H5N8) virus hemagglutinin was undertaken in the Alphafold2 colab program version 1.5.5 (https://colab.research.google.com/github/sokrypton/ColabFold/blob/main/AlphaFold2.ipynb, accessed on 15 November 2023), with the following settings: template mode for detection—PDB100; MSA options—msa_mode = mmseqs2_uniref_env; pair_mode = “unpaired_paired” (“unpaired_paired” = paired sequences of the same species + unpaired MSA); alphafold2_multimer_v3 mode was used for trimer modeling; number of recycles 3. The greedy merging strategy was adopted (merging any taxonomically matching subsets). Models were visualized using RCSB PDB website, Mol*Plagin 3.43.1 3D Viewer tool, (https://www.rcsb.org/3d-view, accessed on 15 November 2023).

### 2.4. Purification of Recombinant Hemagglutinin Protein

Recombinant HA/H5 hemagglutinin was purified by affinity chromatography on a Ni-NTA column (Qiagen, Venlo, The Netherlands). Before sample loading, the culture medium was centrifuged for 15 min at 5000 rpm at 4 °C to remove cellular debris. Thereafter, the supernatant was diluted 1:1 with binding buffer (20 mM Na_2_HPO_4_, 0.5 M NaCl, pH 7.4) and loaded onto the column. The column was washed with wash buffer (20 mM Na_2_HPO_4_, 0.5 M NaCl, 30 mM imidazole, pH 7.4). The target protein was eluted with elution buffer (20 mM Na_2_HPO_4_, 0.5 M NaCl, 500 mM imidazole, pH 7.4). The degree of purification of the target protein was assessed by electrophoresis in 7.5% PAGE followed by fixation and Coomassie G250 staining. Fractions containing the target protein were pooled and dialyzed against PBS (Neofroxx, Einhausen, Germany). Protein samples were then quantified using the Bradford assay.

### 2.5. Gel Permeation Chromatography (GPC)

The molecular weight distribution of the samples was investigated by gel permeation chromatography (GPC) on an Agilent 1200 chromatograph with column PL aquel-OH 40, 300  ×  7.5 mm column (Agilent, Santa Clara, CA, USA) at 30 °C with a refractometric detector. The solvent was PBS (pH = 4.75) aqueous solution and the flow rate was 1 mL/min. The concentrations of solutions were 0.2 wt.%. Agilent GPC data analysis program was used to process the results.

### 2.6. Western Blot Analysis

Western blot analysis was performed using the SNAP i.d. 2.0 system (Millipore, Burlington, MA, USA) according to the manufacturer’s recommendations. Serum of ferret infected with influenza A (H5N8) virus (1:200) (FBRI SRC VB «Vector», Rospotrebnadzor) was used as the primary antibody. Mouse anti-ferret IgG (1:3000) (FBRI SRC VB «Vector», Rospotrebnadzor) and goat anti-mouse IgG-alkaline phosphatase (1:5000) (Sigma) were used as the secondary antibodies. The immune complex was visualized by adding 1-Step™ NBT/BCIP substrate (Thermo Fisher Scientific, Waltham, MA, USA).

### 2.7. Hemagglutination Assay and Hemagglutination Inhibition Assay

The hemagglutination reaction was performed in 96-well round-bottom serological plates according to the method described by Athmaram et al. [16]. Briefly, 50 µL of two-fold serial dilutions of protein and inactivated virus preparations in PBS were added to the wells of the plate. Then 50 µL of 0.5% turkey erythrocytes (FBRI SRC VB “Vector”, Rospotrebnadzor) in PBS were added to each well and incubated for 30 min at room temperature. The HA titer was determined as the highest dilution of the antigen that caused a hemagglutination reaction.

The hemagglutination-inhibition reaction was performed as follows. First, the virus titer was determined in the hemagglutination reaction and the virus was diluted so that 25 µL contained 4 HA units of virus. Then, 25 µL of double dilutions of each serum were prepared in a 96-well round-bottom plate, an equal volume (25 µL) of the prepared standardized virus was added to all wells, and incubated at 22–25 °C for 30 min; an equal volume (50 µL) of turkey erythrocytes was added and incubated at 4 °C for 30–40 min. The serum titer was defined as the highest serum dilution that inhibited hemagglutination.

### 2.8. Laboratory Animals and Immunization Procedures

All applicable international, national, and institutional guidelines for the care and use of animals were followed. Animal experiments were carried out in accordance with the Russian laws and regulations as well as the No. 1 Bioethics Protocol (21 March 2023) of the Vector BioEthics Committee, at Rospotrebnadzor.

Female BALB/c mice (16–18 g) were used for the experiments. Animals were housed according to the standard laboratory conditions with access to food and water. Animals were divided into three groups, which were intramuscularly (i.m.) immunized twice at a 3-week interval. The first group (*n* = 10) was injected with 50 µg of purified recombinant HA/H5 hemagglutinin resuspended in physiological saline in the presence of complete Freund’s adjuvant (the 1st immunization) and incomplete Freund’s adjuvant (the 2nd immunization) (Sigma) in a total volume of 100 µL. The second group (*n* = 10) was injected with 25 μL of β-propiolactone inactivated A/Astrakhan/3212/2020 (H5N8) of 1 × 10^6^ TU/mL also in combination with complete Freund’s adjuvant (the 1st immunization) and incomplete Freund’s adjuvant (the 2nd immunization) (Sigma) in a total volume of 100 μL as a positive control. The third group (*n* = 10) consisted of intact animals. Blood samples were collected from the retro-orbital sinus 14 days after the second immunization. The blood was incubated for 1 h at 37 °C and 2 h at 4 °C, then centrifuged at 7000× *g* for 10 min and serum was collected. Serum was inactivated by heating at 56 °C for 30 min and stored at −20 °C.

### 2.9. Virus Challenge

Manipulations with animals were carried out in compliance with the rules of asepsis and antisepsis.

All studies to evaluate the protective properties of recombinant hemagglutinin conformed to the Sanitary Rules and Regulations (SanPin) 3.3686-21 [23].

Fourteen days after the second immunization, animals were infected by intranasal inoculation of 20 MLD50 of A/Astrakhan/3212/2020 (H5N8) influenza virus diluted in 25 µL under anesthesia with a combination of Zoletil 100 (Virbac, France) and Xyla (Interchemie, Harju maakond, Estonia).

The animals were checked daily for 14 days after infection, and clinical signs as indicators of the disease (activity, appetite) and death were simultaneously monitored. In cases where the mice developed severe conditions incompatible with life, e.g., anorexia, lethargy, an animal was sacrificed. After 14 days of infection, mice were humanely euthanized with CO_2_.

### 2.10. Enzyme-Linked Immunosorbent Assay (ELISA)

Enzyme-linked immunosorbent assay (ELISA) analysis was performed as described by Borgoyakova et al. [24]. Recombinant hemagglutinin of influenza A (H5N8) virus obtained in this work was used as an antigen. Sera of immune animals (dilution from 1:10) were used as the primary antibodies. Goat anti-mouse IgG antibodies conjugated with horseradish peroxidase (HRP) (Sigma) were used as the secondary antibodies. Tetramethylbenzidine (IMTEK) solution was used as a chromogenic substrate. The reaction was stopped with 1 N hydrochloric acid solution. The optical density was measured on a Varioskan LUX Microplate Reader (Thermo Fisher Scientific) at a wavelength of 450 nm. The endpoint titer was determined by the last diluted specimen that gave positive ELISA results.

### 2.11. In Vitro Microneutralization Assay

The *in vitro* microneutralization assay was performed as described by Gross et al. [25]. The animal sera were treated with receptor-destroying enzyme (RDE) and heat inactivated prior to the assay. Each sample was for a final pre-dilution of 1:10. The standardized virus was 200 TCID50/200 µL of viral diluent. Double dilutions of the blood serum were prepared in 200 µL of viral diluent, starting from 1:10; then, 200 µL of the standardized virus were added into each test tube and the tubes were incubated for 1 h at 37 °C, 5% CO_2_. Subsequently, 200 µL suspensions were carried over to the wells of a 96-well plate with MDCK-SIAT1 cell cultures; the cells were incubated for 48 h at 37 °C, 5% CO_2_. After incubation, cells were stained with crystal violet solution (1.3 g of the dye was dissolved in 50 mL of 96% ethyl alcohol, brought to 700 mL with distilled water, and added to 300 mL of 40% formalin solution) and analyzed using an Agilent BioTek Cytation 5 multi-mode cell visualization reader (Thermo Fisher Scientific). Virus neutralization titer was defined as the serum dilution at which 50% of cells survived.

### 2.12. Statistics

Data were analyzed in GraphPad Prism 9.0 software (GraphPad Software Inc.). Results were expressed as a median with a range. Data were analyzed through nonparametric tests. Intergroup differences in immune responses were assessed using nonparametric one-factor Kruskal–Wallis analysis of variance. *p* < 0.05 was considered statistically significant. Comparisons were not statistically significant unless otherwise noted.

## 3. Results

### 3.1. Expression and Characterization of Soluble Recombinant H5N8 Hemagglutinin Protein

In this study, the design of recombinant hemagglutinin was based on the influenza A (H5N8) virus circulating across farm birds in the Stavropol Territory in 2020 (A/turkey/Stavropol/320-01/2020). The following modifications were made to the original amino acid sequence during the design: the transmembrane and cytoplasmic domains were removed [13]; the cleavage site (cleavage site) between HA1 (head domain) and HA2 (stem) PLREKRRRKRG was replaced with PLREKG to preserve the uncleaved protein; a T4 trimerizing domain was added at the C-terminus to stabilize the trimer structure [26,27]; and a poly-His-tag was added for subsequent purification [28] (Figure 1a). A model of the engineered hemagglutinin trimer is shown in Figure 1b. 

Following that, the codon composition of the gene encoding the designed hemagglutinin was optimized to support its efficient expression in mammalian cells. The recombinant hemagglutinin gene was obtained by chemical synthesis and cloned into the plasmid vector pVL3. As a result, the plasmid pVL3-HA/H5 was obtained; the genetic map is shown in Figure 1c.

A stable producer of recombinant hemagglutinin of the influenza A virus (H5N8) was created based on the CHO-K1 cell line. For preparative production of the target protein, CHO-K1-HA/H5 cells were cultured in roller bottles with subsequent purification of the recombinant protein by affinity chromatography and dialyzed against PBS. The solution remained clear following the dialysis of the protein preparation obtained after chromatography, which indicated that the recombinant protein remained in the soluble form.

PAGE electrophoresis revealed that in terms of electrophoretic mobility, the purified protein corresponded to the theoretically calculated molecular mass of ~70 kDa for the monomer and ~210 kDa for the trimer (Figure 2a).

Western blot analysis showed that HA monomers and trimers were recognized by ferret serum infected with influenza A (H5N8) virus under both denaturing and native conditions, respectively (Figure 2b).

The purified HA/H5 was also analyzed by gel permeation chromatography for its ability to form higher order oligomers, particularly trimers. The gel filtration results confirmed the PAGE data that the protein was in solution in both monomeric and trimeric forms (Figure 2c).

We also tested recombinant HA/H5 in a hemagglutination assay and found that the purified protein was able to agglutinate turkey red blood cells (Figure 2d). This may indicate that recombinant HA/H5 folds into its native structure and retains the ability to bind sialic acid receptors on erythrocytes, causing agglutination.

Thus, recombinant hemagglutinin of at least 90% purity was obtained. Protein production was about 30 mg per 0.5 L of culture medium.

### 3.2. Evaluation of Immunogenic and Protective Properties

To analyze the immunogenicity of recombinant hemagglutinin HA/H5, Balb/c mice were immunized. The animals were divided into three groups: Group 1 was injected with recombinant hemagglutinin HA/H5; Group 2 was injected with β-propiolactone inactivated virus A/Astrakhan/3212/(H5N8)/CE/E1; Group 3 was a control group of intact animals (Figure 3a). ELISA test of immune sera showed that recombinant hemagglutinin HA/H5 was able to induce a specific humoral immune response (the average antibody titer was 505,913). Specific antibodies were also detected in the group of animals immunized with inactivated virus (the average titer was 66,825) (Figure 3b).

The immune sera were tested for *in vitro* neutralizing activity against influenza virus H5N8 using MDCK cell culture. As a result, it was found that the sera of animals immunized with both recombinant hemagglutinin HA/H5 and inactivated virus A/Astrakhan/3212/(H5N8)/CE/E1 were able to neutralize the live influenza virus strain A/turkey/Stavropol/320-01/2020 (Figure 3c), with average titers of 2560 and 1280, respectively. No neutralizing activity was detected in the mouse serum of the control group.

The HI antibody titers of the immune sera were determined in the hemagglutination-inhibition reaction. As a result, the sera from animals immunized with recombinant HA/H5 haemagglutinin were found to interact with the homologous strain and the recommended past vaccine strains (Table 1) and cause hemagglutination inhibition. Almost no reaction was observed against A/chicken/Khabarovsk/24-1V/2022 (H5N1) strain 2.3.4.4.b. The results are in agreement with the data of Marchenko et al. [21].

Fourteen days after the 2nd immunization, mice were infected intranasally with 20 MLD50 of A/Astrakhan/3212/2020 (H5N8) influenza virus. It was found that immunization with both recombinant HA/H5 hemagglutinin and inactivated virus gave 100% protection against H5N8 virus infection (Figure 4).

## 4. Discussion

Avian influenza viruses pose a threat not only to wild and farm birds, but also to mammals, particularly humans, due to their mutational variability, recombination, and the possibility of interspecies transmission [29].

In recent years, the highly pathogenic viruses H5N1, H5N6, and H5N8 have continued to evolve in wild and domestic birds and have caused numerous human cases [29]. The ongoing threat from various subtypes of avian influenza viruses highlights the importance of not only surveillance of their natural reservoirs, but also the implementation of vaccine prophylaxis to prevent the spread of these influenza viruses.

Hemagglutinin (HA) is the major viral protein recognized by the immune system and is subsequently the primary target for the development of vaccines against relevant influenza viruses [14,30,31,32]. HA-based vaccines are emerging as a good alternative to egg-based influenza vaccine [33,34]. To ensure HA antigenicity, it should be in a trimerized form and undergo post-translational modifications (e.g., glycosylation) that can be provided by a eukaryotic expression system using mammalian cell cultures [35,36].

In this study, a stable producer of recombinant hemagglutinin (HA) of the influenza A virus (H5N8) was created based on the CHO-K1 cell line.

Since HA is active as a trimer on the virus surface, HA trimerization is necessary to ensure the correct immunogenic conformation. For these purposes, we used the following modifications: the transmembrane and cytoplasmic domains were removed [13] and a T4 bacteriophage fibritin trimerization domain was added [26,27], while the cleavage site between HA1 and HA2 PLREKRRRKRG was replaced with PQRETRG to preserve the uncleaved protein. It is known that removing or modifying a cleavage site can help stabilize the structure in the pre-fusion conformation, as stabilization of the pre-fusion conformations of fusion proteins has been shown to be a key success factor in the induction of an effective immune response [32,37]. The formation of trimers was confirmed by gel permeation chromatography (Figure 2c).

After purification, preparative amounts of recombinant HA/H5 hemagglutinin with a purity of at least 90% were obtained (Figure 2a). Using immunoblot, it was shown that the recombinant hemagglutinin was recognized by the serum of ferrets infected with A (H5N8) influenza virus, which may indicate that its antigenic properties were retained (Figure 2b). The functionality of the recombinant hemagglutinin was also confirmed in a hemagglutination test (Figure 2d).

Immunogenicity analysis of the HA protein showed that double immunization of mice with recombinant HA/H5 hemagglutinin elicited a specific humoral immune response with the formation of neutralizing antibodies (Figure 3).

To evaluate the protective efficacy of recombinant HA/H5 hemagglutinin, mice were infected with homologous A/Astrakhan/3212/2020 (H5N8) influenza virus at a dose of 20 (MLD50) on the 35th day after the start of the experiment. The protective efficacy of recombinant HA/H5 hemagglutinin against lethal A/Astrakhan/3212/2020 (H5N8) influenza virus infection reached 100% (Figure 4).

The results presented in this study and data presented by other authors [12,13,16,17,18,19,20] indicate that recombinant hemagglutinins are capable of protecting animals from viral infection and can be used as a component of a vaccine against influenza A virus.

The recombinant HA can be used to investigate both structural features and biological properties of influenza A (H5N8) virus proteins responsible for virus entry into permissive cells.

## 5. Conclusions

Thus, in this study CHO-K1 cells, producers of recombinant hemagglutinin of A/turkey/Stavropol/320-01/2020 (H5N8) influenza virus, were obtained.

Purified recombinant hemagglutinin (HA) of influenza A (H5N8) virus is highly immunogenic and induces protective immune responses against lethal challenge with the homologous A/Astrakhan/3212/2020 (H5N8) influenza virus.

The resulting protein may be useful in serologic diagnostics. It can also be a vaccine component enabling effective control of A (H5N8) influenza virus.

## Figures and Tables

**Figure 1 vaccines-12-00143-f001:**
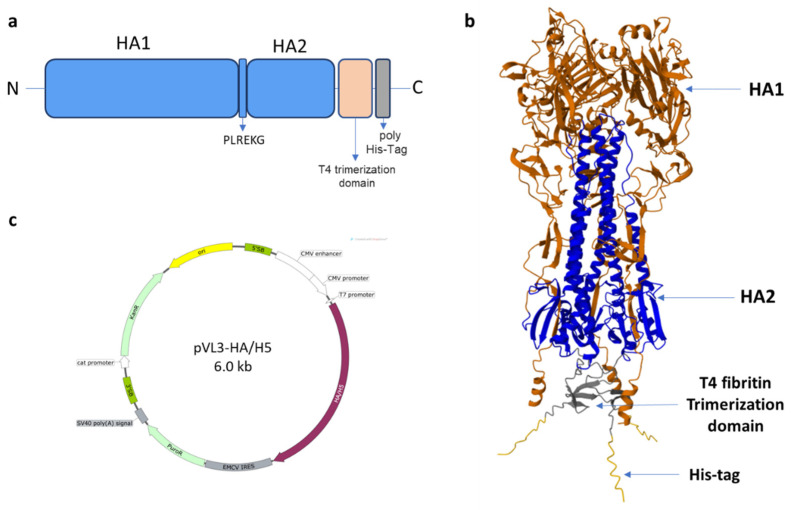
(**a**) Scheme of the recombinant hemagglutinin design; (**b**) the trimeric HA/H5 protein model (The structural modeling of protein was performed with AlphaFold2 using ColabFold as described in Section 2); (**c**) vector map of the recombinant construct, designed using SnapGene 3.2.1.

**Figure 2 vaccines-12-00143-f002:**
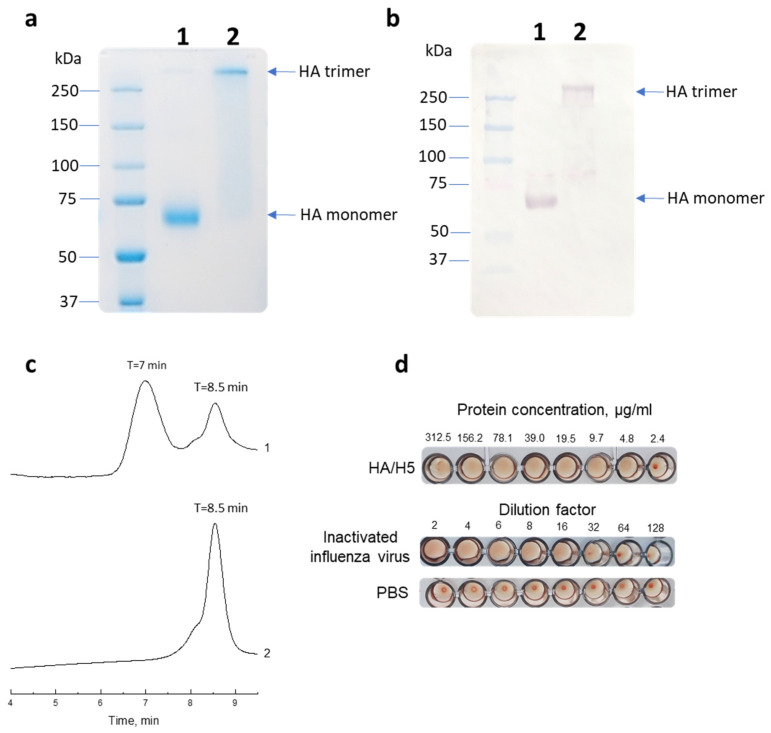
(**a**) Electrophoretic mobility of HA/H5 under denaturing (1) and under non-denaturing conditions (2). (**b**) Western blot analysis of the recombinant HA/H5 using ferret H5N8 serum under denaturing (1) and under non-denaturing conditions (2). (**c**) Gel chromatogram of purified HA/H5 protein (1) and BSA (2). (**d**) Hemagglutination assay using the HA/H5 recombinant protein. Inactivated A/turkey/Stavropol/320-01/2020 (H5N8) influenza virus was used as a positive control. PBS was used as a negative control.

**Figure 3 vaccines-12-00143-f003:**
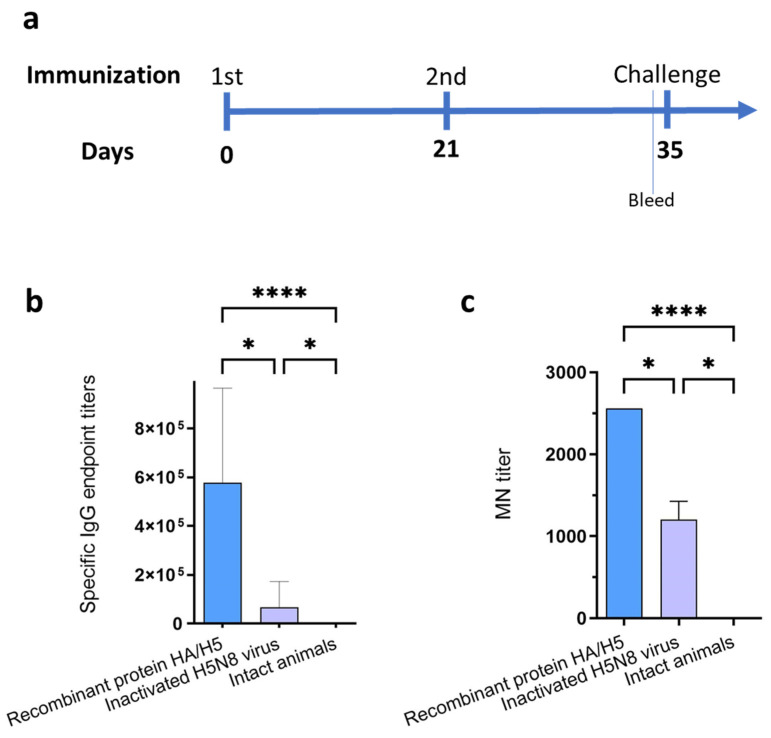
(**a**) Scheme of immunization and challenge. (**b**) Titers of hemagglutinin-specific antibodies detected in immune sera by ELISA. Recombinant hemagglutinin HA/H5 was used as an antigen. Specific IgG endpoint titers are marked on the ordinate axis. (**c**) Virus neutralizing activity of immune sera against influenza A/turkey/Stavropol/320-01/2020 (H5N8) virus. MN (microneutralization) titer is marked on the ordinate axis. Data are presented as median with range. Statistics were calculated using the Kruskal–Wallis test (* *p* < 0.05; **** *p* < 0.0001).

**Figure 4 vaccines-12-00143-f004:**
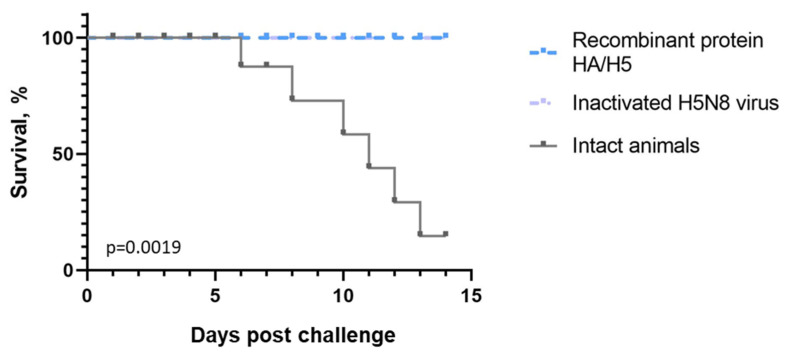
Survival curves of immunized animals after infection with influenza A/Astrakhan/3212/2020 (H5N8) strain. On the ordinate axis—% of surviving animals, on the abscissa axis—days from the moment of infection. The differences between survival rates in the studied groups are statistically significant according to the Mantel–Cox test (*p* = 0.0019).

**Table 1 vaccines-12-00143-t001:** Results of hemagglutination-inhibition reaction with A (H5Nx) viruses, clade 2.3.4.4.

Serum ID	Viruses				
	A/Dalmatian Pelican/Astrakhan/213-2V/2022 (H5N1) 2.3.4.4.b	A/Chicken/Khabarovsk/24-1V/2022 (H5N1) 2.3.4.4.b	A/Astrakhan/3212/2020 (H5N8) 2.3.4.4b	A/Gyrfalcon/Washington/41088-6/2014 (H5N8) 2.3.4.4c	A/Chicken/Vietnam/NCVD-15A59/2015 (H5N6) 2.3.4.4f
1	≤10	≤10	≤10	≤10	≤10
2	≤10	≤10	≤10	≤10	≤10
3	≤10	≤10	≤10	≤10	≤10
4	320	≤10	80	80	80
5	320	≤10	80	80	80
6	320	≤10	80	80	80
7	≤10	≤10	≤10	≤10	≤10
8	80	≤10	80	≤10	≤10
9	80	≤10	80	80	80
10	≤10	≤10	≤10	≤10	≤10

Note: For intact sera and sera from animals immunized with inactivated virus, the HI antibody titers were ≤10.

## Data Availability

The data presented in this study are available on request from the corresponding author.

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
