# Peer review of "Immunogenic and Protective Properties of Recombinant Hemagglutinin of Influenza A (H5N8) Virus"

_vaccines, 2024, doi:10.3390/vaccines12020143_

Round 1
Reviewer 1 Report
Comments and Suggestions for Authors
Comments
The endpoint titer is defined as the reciprocal of the highest dilution that gives a reading above the cutoff. The titer should be written as eg 2560 and not 1:2560
It would be great to know
(i) the HA titer of the purified rH5. It will also confirm the functionality and antigenicity of the rH5 trimer as a HA protein, although the antigenicity is evidenced by the Western blotting native conditions and immunoblots.
(ii) the solubility of the rH5 trimeric protein and the yield of the rH5.
Author Response
We would like to thank the reviewer for careful and thorough reading of our manuscript and for the thoughtful comments and constructive suggestions, which help to improve its quality. Our response follows.
Corrections to the text are highlighted in green.
The endpoint titer is defined as the reciprocal of the highest dilution that gives a reading above the cutoff. The titer should be written as eg 2560 and not 1:2560
The correction has been made. Please see page 1, line 21, page 7, lines 291, 292, 298 of the revised manuscript.
It would be great to know
(i) the HA titer of the purified rH5. It will also confirm the functionality and antigenicity of the rH5 trimer as a HA protein, although the antigenicity is evidenced by the Western blotting native conditions and immunoblots.
The addition has been made. Please see page 6 of the revised manuscript, lines 271-274, and page 7 of the revised manuscript, lines 281-283.
A description of the methodology is provided in subparagraph 2.7 (page 4).
(ii) the solubility of the rH5 trimeric protein and the yield of the rH5.
The correction has been made. Please see page 6 of the revised manuscript, lines 258-260 and lines 275-276.
Reviewer 2 Report
Comments and Suggestions for Authors
This manuscript provides data on the immunogenicity and protective efficacy of a recombinant hemagglutinin vaccine. However, the significance of the vaccine and the necessary background of the manuscript are insufficiently highlighted. The manuscript should effectively persuade readers that the vaccine candidate is useful and promising for a possible pandemic threat. There are some major and minor problems to be addressed before publication.
Majors
1. Please provide more information about the threat of clade 2.3.4.4 viruses.
a. The threat posed by clade 2.3.4.4 viruses is not described in the introduction. Avian influenza viruses continue to challenge both animal and human health. A paper (ID: 36458831) summarized the impact of H5 viruses on the poultry industry and public health. Please include data regarding human infections and poultry losses.
b. Studies (PMID: 35380505) indicate that 2.3.4.4 viruses are continually evolving, resulting in 8 subclades (2.3.4.4a-2.3.4.4h) and several antigenic groups. Notably, clade 2.3.4.4b viruses are prevalent worldwide. Authors should reference these and other papers to describe the epidemiology of clade 2.3.4.4 viruses and explain why H5 viruses pose a significant public health threat.
2. The antigenicity of clade 2.3.4.4 viruses varies. Will the vaccine in this study offer protective efficacy against other 2.3.4.4 subclades? If possible, authors should provide data on the vaccine's protective efficacy against other H5 viruses (clade 2.3.4.4a-h) in animals, or at least the cross-reactive HI titers.
Minors
1. Lines 59-70: Please provide the sequence IDs for the viruses mentioned. Additionally, more information on the bacteria and cell cultures is required, such as the reagents used for culturing.
2. Lines 204-219: The method should have included details about the experiments, please simplify these paragraphs.
3. Figure 2: The gel chromatogram is mentioned but not described in the methodology. It is advisable to describe it in an appropriate section.
4. Lines 274-276: Clarify the significance of the uncleaved protein in inducing an immune response.
5. Lines 285-289: This section merely describes the results. The subunit vaccine provides 100% protection against the homologous virus, but what about its efficacy against other 2.3.4.4 subclades?
Comments on the Quality of English Language
Minor editing of English language required
Author Response
We would like to thank the reviewer for careful and thorough reading of our manuscript and for the thoughtful comments and constructive suggestions, which help to improve its quality. Our response follows.
Corrections to the text are highlighted in green.
Majors
- Please provide more information about the threat of clade 2.3.4.4 viruses.
The correction has been made. Please see page 1 of the revised manuscript, lines 30-39.
a. The threat posed by clade 2.3.4.4 viruses is not described in the introduction. Avian influenza viruses continue to challenge both animal and human health. A paper (ID: 36458831) summarized the impact of H5 viruses on the poultry industry and public health. Please include data regarding human infections and poultry losses.
The correction has been made. Please see page 1 of the revised manuscript, lines 30-39.
b. Studies (PMID: 35380505) indicate that 2.3.4.4 viruses are continually evolving, resulting in 8 subclades (2.3.4.4a-2.3.4.4h) and several antigenic groups. Notably, clade 2.3.4.4b viruses are prevalent worldwide. Authors should reference these and other papers to describe the epidemiology of clade 2.3.4.4 viruses and explain why H5 viruses pose a significant public health threat.
The correction has been made. Please see page 1 of the revised manuscript, lines 40-48.
- The antigenicity of clade 2.3.4.4 viruses varies. Will the vaccine in this study offer protective efficacy against other 2.3.4.4 subclades? If possible, authors should provide data on the vaccine's protective efficacy against other H5 viruses (clade 2.3.4.4a-h) in animals, or at least the cross-reactive HI titers.
We agree that the vaccine under development should also be able to elicit a protective immune response against other 2.3.4.4 subclades. However, this question can only be answered after a proper experiment has been performed. The results presented in the manuscript are the outcome of the first stage of the study of the obtained recombinant haemagglutinin. Since the experiments to study the protective efficacy are labor-intensive and time-consuming, we performed the analysis on only one virus. Further, we plan to evaluate the protective properties of the recombinant haemagglutinin against other 2.3.4.4 subclades.
An appropriate experiment was conducted to determine the cross-reactive HI titers. The results are presented on pages 8-9, lines 307-327.
Minors
- Lines 59-70: Please provide the sequence IDs for the viruses mentioned. Additionally, more information on the bacteria and cell cultures is required, such as the reagents used for culturing.
The correction has been made. Please see page 2 of the revised manuscript, subparagraph 2.1.
- Lines 204-219: The method should have included details about the experiments, please simplify these paragraphs.
The correction has been made. Please see page 6 of the revised manuscript, lines 255-261.
- Figure 2: The gel chromatogram is mentioned but not described in the methodology. It is advisable to describe it in an appropriate section.
The correction has been made. Please see page 3 of the revised manuscript, subparagraph 2.5.
- Lines 274-276: Clarify the significance of the uncleaved protein in inducing an immune response.
The correction has been made. Please see page 10 of the revised manuscript, lines 360-364.
«It is known that removal or modification of the cleavage site can help stabilise the structure in the pre-fusion conformation, as stabilisation of the pre-fusion conformations of fusion proteins has been shown to be a key success factor in the induction of an effective immune response [Weldon WC, Wang B-Z, Martin MP, Koutsonanos DG, Skountzou I, Compans RW (2010) Enhanced Immunogenicity of Stabilized Trimeric Soluble Influenza Hemagglutinin. PLoS ONE 5(9): e12466. https://doi.org/10.1371/journal.pone.0012466; Milder, F.J.; Jongeneelen, M.; Ritschel, T.; Bouchier, P.; Bisschop, I.J.; de Man, M.; Veldman, D.; Le, L.; Kaufmann, B.; Bakkers, M.J.G.; Juraszek, J.; Brandenburg, B.; Langedijk, J.P. Universal stabilization of the influenza hemag-glutinin by structure-based redesign of the pH switch regions. Proceedings of the National Academy of Sciences 2022, 119, e2115379119. doi: 10.1073/pnas.2115379119]».
- Lines 285-289: This section merely describes the results. The subunit vaccine provides 100% protection against the homologous virus, but what about its efficacy against other 2.3.4.4 subclades?
We agree that the vaccine under development should also be able to elicit a protective immune response against other 2.3.4.4 subclades. However, this question can only be answered after a proper experiment has been performed. The results presented in the manuscript are the outcome of the first stage of the study of the obtained recombinant haemagglutinin. Since the experiments to study the protective efficacy are labor-intensive and time-consuming, we performed the analysis on only one virus. Further, we plan to evaluate the protective properties of the recombinant haemagglutinin against other 2.3.4.4 subclades.
Reviewer 3 Report
Comments and Suggestions for Authors
The manuscript “Immunogenic and protective properties of recombinant hemagglutinin of influenza A (H5N8) virus” by Rudometova et al. discusses the production and immunogenicity of two potential vaccine candidates: recombinant H5N8 protein and inactivated H5N8 virus. This study has shown that both vaccine candidates have protected the mice after challenge with homologous A/Astrakhan/3212/2020 (H5N8).
The production of recombinant haemagglutinin protein using CHO-K1 cells will speed up the mass production of rH5 in a pandemic situation.
1. Overall, the introduction section and material methods need more detailed information.
2. In the introduction section, provide some background on not using rH5 from H5N1 rather H5N8 for the production of rH5 protein, as both have the same haemagglutinin subtype.
3. What is the reason behind using (H5N8) virus (A/turkey/Stavropol/320-01/2020) for rH5 protein and not A/Astrakhan/3212/2020 (H5N8)? Also, for the challenge study, A/Astrakhan/3212/2020 (H5N8) was used and not (A/turkey/Stavropol/320-01/2020.
4. Lines 120–121 mention the source of ferret serum.
5. Line 133, how was the dose of 50 ug selected? any previous reference related to the experimental protocol.
6. Line 156, sentence ‘described in [14] in our modification’.
7. Provide brief MN assay methods, mention initial dilution, etc.
8. Why was no shedding of the influenza virus studied in challenge studies?
9. In result section 3.2, what could be the reason why there are so many differences in Avg ELISA titers for recombinant and inactivated vaccines (almost 7.5 times), although in microneutralization tests there is only a 2-fold difference (1:2650 and 1:1280)?
10. Line 240, Are you sure the titer is 1:2650 or 1:2560 (i.e., a 2-fold dilution of 1:1280)?
11. Why was the HI test not conducted to study the serum titer?
Author Response
We would like to thank the reviewer for careful and thorough reading of our manuscript and for the thoughtful comments and constructive suggestions, which help to improve its quality. Our response follows.
Corrections to the text are highlighted in green.
- Overall, the introduction section and material methods need more detailed information.
The correction has been made. Please see the "Introduction" and "Materials and Methods" sections.
- In the introduction section, provide some background on not using rH5 from H5N1 rather H5N8 for the production of rH5 protein, as both have the same haemagglutinin subtype.
The correction has been made. Please see the "Introduction" sections, lines 56-68.
- What is the reason behind using (H5N8) virus (A/turkey/Stavropol/320-01/2020) for rH5 protein and not A/Astrakhan/3212/2020 (H5N8)? Also, for the challenge study, A/Astrakhan/3212/2020 (H5N8) was used and not (A/turkey/Stavropol/320-01/2020.
Virus A/Astrakhan/3212/2020 (H5N8) was isolated from humans [Pyankova, O.G.; Susloparov, I.M.; Moiseeva, A.A.; Kolosova, N.P.; Onkhonova, G.S.; Danilenko, A.V.; Vakalova, E.V.; Shendo, G.L.; Nekeshina, N.N.; Noskova L.N.; Demina, J.V.; Frolova, N.V.; Gavrilova, E.V.; Maksyutov, R.A.; Ryzhikov, A.B. Isolation of clade 2.3.4.4b A(H5N8), a highly pathogenic avian influenza virus, from a worker during an outbreak on a poultry farm, Russia, December 2020. Eurosurveillance 2021, 26, 2100439. doi: 10.2807/1560-7917.ES.2021.26.24.2100439.], and was selected by WHO as a candidate vaccine strain in the event of a human outbreak caused by antigenically related strains of the virus [https://cdn.who.int/media/docs/default-source/influenza/who-influenza-recommendations/vcm-northern-hemisphere-recommendation-2022-2023/202203_zoonotic_vaccinevirusupdate.pdf]. The A/turkey/Stavropol/320-01/2020 (H5N8) virus was isolated during the same outbreak as the previous strain, but from turkey. Therefore, both strains were used in testing our vaccine.
- Lines 120–121 mention the source of ferret serum.
The correction has been made. Please see page 4 of the revised manuscript, line 153.
- Line 133, how was the dose of 50 ug selected? any previous reference related to the experimental protocol.
We used a protein dose of 50 μg/mouse because for mice, if pure soluble protein antigen is used and is available in large quantities, the general recommendation is a dose of 50-100 μg plus adjuvant given intramuscularly at each immunization [Greenfield EA. Standard Immunization of Mice, Rats, and Hamsters. Cold Spring Harb Protoc. 2020 Mar 2;2020(3):100297. doi: 10.1101/pdb.prot100297. PMID: 32123014.].
- Line 156, sentence ‘described in [14] in our modification’.
The correction has been made. Please see page 5 of the revised manuscript, subparagraph 2.11.
- Provide brief MN assay methods, mention initial dilution, etc.
The correction has been made. Please see page 5 of the revised manuscript, subparagraph 2.11.
- Why was no shedding of the influenza virus studied in challenge studies?
We agree that an experiment to isolate the virus from immunized animals and analyze the transmissibility of the virus is important for assessing the effectiveness of the vaccine. We will try to carry out this analysis in the next stages of the study of our protein. At the first stage of the study, it was important for us to evaluate the immunogenic and protective properties of the recombinant protein.
- In result section 3.2, what could be the reason why there are so many differences in Avg ELISA titers for recombinant and inactivated vaccines (almost 7.5 times), although in microneutralization tests there is only a 2-fold difference (1:2650 and 1:1280)?
In the neutralization analysis, the 2-fold difference may be due to the fact that during immunization with an inactivated virus, antibodies are formed not only to hemagglutinin, but, as is known, also to neuraminidase, which also contributes to the neutralization of the virus.
In ELISA, the difference is explained by the fact that the level of specific antibodies specifically to hemagglutinin was measured, and if we recalculate the dose of inactivated virus administered to hemagglutinin (<50 μg), therefore we observe a 7.5-fold difference in ELISA.
- Line 240, Are you sure the titer is 1:2650 or 1:2560 (i.e., a 2-fold dilution of 1:1280)?
We apologize for our error. The correction has been made. Please see page 7 of the revised manuscript, line 298.
- Why was the HI test not conducted to study the serum titer?
An appropriate experiment was conducted to determine the cross-reactive HI titers. The results are presented on pages 8-9, lines 307-327.
Reviewer 4 Report
Comments and Suggestions for Authors
In their article” Immunogenic and protective properties of recombinant hemagglutinin of influenza A (H5N8) virus “ the authors present a study, whose aim is to present an antigen candidates for the development of subunit vaccines against influenza A (H5N8) virus with suitable immunogenicity and protection efficacy.
I think the title is better to change, for example, “Probable or future candidate for development of subunit vaccines against influenza A” or something else. My idea is that your study is trying to find a future candidate for a vaccine, and it is good to be mentioned in the title.
Introduction. Line 48- HAs, it is better to write the whole name, not the abbreviation.
Materials and Methods.
2.1. You listed the used viruses, but you need to write how they were prepared for the experiments.
2.2. Line 78-“kindly provided” it is not proper. Add Acknowledgements at the end of the article and there you can thank everybody, who helped you. Here you write, for example, the vector is provided by “Vector”…..
2.9. The first sentence, it is better to write … was performed as described by Gross et all., 2017 (14). Revise the second sentence, “The only difference was in the method of visualization of the final result: ……“ , for example, “We made a modification concerning visualization….”
2.7. and 2.10. Could be united in one point, or have to be one after another.
Results. Revise this part. Here should have only results, no explanation of methods.
Discussion. This part needs major revision. You have to do a discussion of your results. What results do you receive, compared with other studies?
Lines 269-270 You should delete this sentence.
Author Response
We would like to thank the reviewer for careful and thorough reading of our manuscript and for the thoughtful comments and constructive suggestions, which help to improve its quality. Our response follows.
Corrections to the text are highlighted in green.
In their article” Immunogenic and protective properties of recombinant hemagglutinin of influenza A (H5N8) virus “ the authors present a study, whose aim is to present an antigen candidates for the development of subunit vaccines against influenza A (H5N8) virus with suitable immunogenicity and protection efficacy.
I think the title is better to change, for example, “Probable or future candidate for development of subunit vaccines against influenza A” or something else. My idea is that your study is trying to find a future candidate for a vaccine, and it is good to be mentioned in the title.
Dear Reviewer,
thank you for your suggestion to change the title of the article. However, we decided to leave the previous title of the article, since the results presented in the manuscript are the result of the first stage of studying the obtained recombinant hemagglutinin, where we showed that the resulting protein has immunogenic and protective properties, and after extensive research can be used as component of the influenza A (H5N8) virus vaccine.
Introduction. Line 48- HAs, it is better to write the whole name, not the abbreviation.
The correction has been made. Please see page 2 of the revised manuscript, line 68.
Materials and Methods.
2.1. You listed the used viruses, but you need to write how they were prepared for the experiments.
The correction has been made. Please see page 2 of the revised manuscript, subparagraph 2.1.
2.2. Line 78-“kindly provided” it is not proper. Add Acknowledgements at the end of the article and there you can thank everybody, who helped you. Here you write, for example, the vector is provided by “Vector”…..
The correction has been made. Please see page 11 of the revised manuscript, the Acknowledgments section.
2.9. The first sentence, it is better to write … was performed as described by Gross et all., 2017 (14). Revise the second sentence, “The only difference was in the method of visualization of the final result: ……“ , for example, “We made a modification concerning visualization….”
The correction has been made. Please see page 5 of the revised manuscript, subparagraph 2.11.
2.7. and 2.10. Could be united in one point, or have to be one after another.
We have placed the sections "Laboratory Animals and Immunization Procedures" and "Virus Challenge" one after the other.
Results. Revise this part. Here should have only results, no explanation of methods.
We have revised the text to address the reviewer’s concern and we hope that it is clearer now.
Discussion. This part needs major revision. You have to do a discussion of your results. What results do you receive, compared with other studies?
We have revised the text to address the reviewer’s concern and we hope that it is clearer now.
Lines 269-270 You should delete this sentence.
We have removed this sentence.
Round 2
Reviewer 2 Report
Comments and Suggestions for Authors
It's suitable for publication now.
Author Response
The authors thank the reviewer for his positive assessment of the manuscript after the changes were made.
Reviewer 3 Report
Comments and Suggestions for Authors
The authors have adequately addressed the queries in revised manuscripts but in revised manuscripts, the addition of the HI titer data (Table 1) has raised some major questions related to the immunogenicity of this recombinant H5 protein.
I have following questions and concerns about rev2 manuscripts.
1. Lines 30-32 provide a reference for the outbreak data
2. Lines 51 and 54 the references (2,8,9,10) are not matching with the updated list of references. For example, in rev 1; reference 10 is Allen et.al., rev 2 it is Chen et. al, and ref 10 corresponds to no. 16 of rev 2. Please go through the manuscripts and correct the numbers.
3. Subsection 2.7 HA, HI cites some previous references and also add 1-2 sentences about calculation of the HA titer in line 162.
4. Please also mention whether the serum sample was initially pre diluted to 1:5 and then two-fold diluted, as you have mentioned in table 1 <10 HI titer.
5. It is better to write two-fold in place of double dilution and the highest dilution of the last serum dilution (lines 165 and 169).
6. In MNT, ELISA and HI test serum samples were tested in single, duplicate or triplicate.
7. Line 216: remove extra ‘l’ from et al.
8. Line 218: heat inactivated instead of inactivate.
9. Line 268 PAGE and not PAAG.
10. Table 1; serum ids 1, 2, 3, 7 and 10 (5 out of 10 mice) don’t’ show any HI activity against any virus, so this has raised the question of the immunogenicity of rH5 protein. Even with challenge virus, these 5 serum ids are not showing any HI antibodies.
11. Please clarify why these 5 serum ids have less than 10 HI antibody titer.
12. If you have data, provide the MN titer and ELISA titer for all the 10 serum samples to see whether same pattern was observed in these 5 serum ids which have less than 10 HI titer in supplementary table.
13. If you have HI titer with A/turkey/Stavropol/320-01/2020, add into the Table 1 as separate column to show HI activity of these 10 serum samples against rH5 protein.
14. Rephrase line 307 to 311 for clarity.
15. In the discussion section, the sentences are not flowing and it appears as repetition of the result section, try to rearrange and rewrite it for fluency.
Comments on the Quality of English Language
There is need to re check and edited by some native English speakers for grammar and fluency of the sentences throughout the manuscript.
Author Response
We would like to thank the reviewer for careful and thorough reading of our manuscript and for the thoughtful comments and constructive suggestions, which help to improve its quality. The English language of the submitted version of the article was also checked. Our response follows.
Corrections to the text are highlighted in red.
- Lines 30-32 provide a reference for the outbreak data
The correction has been made. Please see the "Introduction" sections, lines 31-32.
- Lines 51 and 54 the references (2,8,9,10) are not matching with the updated list of references. For example, in rev 1; reference 10 is Allen et.al., rev 2 it is Chen et. al, and ref 10 corresponds to no. 16 of rev 2. Please go through the manuscripts and correct the numbers.
We checked the reference list. Since we added new references in the first paragraph, the reference numbers of previous cited works have therefore been changed.
- Subsection 2.7 HA, HI cites some previous references and also add 1-2 sentences about calculation of the HA titer in line 162.
The correction has been made. Please see the Subsection 2.7.
- Please also mention whether the serum sample was initially pre diluted to 1:5 and then two-fold diluted, as you have mentioned in table 1 <10 HI titer.
Undiluted serum was treated with RDE (1 part serum plus 3 parts RDE), incubated for 18 hours at 37°C, then 30 minutes at 56°C (to inactivate RDE), after which 6 parts phosphate-buffered saline was added. As a result, the original serum became diluted 10 times.
- It is better to write two-fold in place of double dilution and the highest dilution of the last serum dilution (lines 165 and 169).
The correction has been made. Please see the Subsection 2.7, lines 158, 161-162, 169.
- In MNT, ELISA and HI test serum samples were tested in single, duplicate or triplicate.
MNT, ELISA and HI tests were performed in duplicate.
- Line 216: remove extra ‘l’ from et al.
The correction has been made. Please see the Subsection 2.11, line 214.
- Line 218: heat inactivated instead of inactivate.
The correction has been made. Please see the Subsection 2.11, lines 215-216.
- Line 268 PAGE and not PAAG.
The correction has been made. Please see page 7 of the revised manuscript, line 267.
- Table 1; serum ids 1, 2, 3, 7 and 10 (5 out of 10 mice) don’t’ show any HI activity against any virus, so this has raised the question of the immunogenicity of rH5 protein. Even with challenge virus, these 5 serum ids are not showing any HI antibodies.
In HI test with antigens A/H5 and A/H7 and mammalian sera, the best results are obtained if horse erythrocytes are used. When using turkey erythrocytes, the titers are significantly lower (5-10 times). Unfortunately, we did not have horse erythrocytes, so in the first version of the article we did not present the results of HI test. We believe that significant titers of neutralizing antibodies fully compensate for the poorly informative results of HI test.
- Please clarify why these 5 serum ids have less than 10 HI antibody titer.
In HI test with antigens A/H5 and A/H7 and mammalian sera, the best results are obtained if horse erythrocytes are used. When using turkey erythrocytes, the titers are significantly lower (5-10 times). Unfortunately, we did not have horse erythrocytes, so in the first version of the article we did not present the results of HI test. We believe that significant titers of neutralizing antibodies fully compensate for the poorly informative results of HI test.
- If you have data, provide the MN titer and ELISA titer for all the 10 serum samples to see whether same pattern was observed in these 5 serum ids which have less than 10 HI titer in supplementary table.
We have added the supplementary materials that presents the MN and ELISA titers for each of the 10 sera.
Group 1 – recombinant hemagglutinin HA/H5
Serum ID |
ELISA titer |
MN titer |
1 |
109350 |
2560 |
2 |
328050 |
2560 |
3 |
1350 |
160 |
4 |
984150 |
2560 |
5 |
984150 |
2560 |
6 |
328050 |
2560 |
7 |
328050 |
2560 |
8 |
984150 |
2560 |
9 |
328050 |
2560 |
10 |
328050 |
2560 |
Group 2 – β-propiolactone inactivated virus А/Astrakhan/3212/(H5N8)/CE/E1
Serum ID |
ELISA titer |
MN titer |
1 |
36450 |
1280 |
2 |
328050 |
1280 |
3 |
36450 |
1280 |
4 |
36450 |
1280 |
5 |
36450 |
1280 |
6 |
12150 |
1280 |
7 |
36450 |
1280 |
8 |
36450 |
1280 |
9 |
36450 |
640 |
10 |
36450 |
1280 |
Group 3 – intact animals
For intact animals, no specific titers were detected in MN and ELISA.
- If you have HI titer with A/turkey/Stavropol/320-01/2020, add into the Table 1 as separate column to show HI activity of these 10 serum samples against rH5 protein.
In the HI test against influenza virus A/turkey/Stavropol/320-01/2020, all sera showed a negative result.
- Rephrase line 307 to 311 for clarity.
The correction has been made. Please see page 8 of the revised manuscript, lines 305-309.
- In the discussion section, the sentences are not flowing and it appears as repetition of the result section, try to rearrange and rewrite it for fluency.
We have revised the Discussion section to address the reviewer’s concern and we hope that it is clearer now.
We would like to thank the referee again for taking the time to review our manuscript
Round 3
Reviewer 3 Report
Comments and Suggestions for Authors
Authors have satisfactorily answered my queries and the manuscript is much improved.